# Differential Photosensitivity of Fibroblasts Obtained from Normal Skin and Hypertrophic Scar Tissues

**DOI:** 10.3390/ijms25042126

**Published:** 2024-02-09

**Authors:** Junya Kusumoto, Masaya Akashi, Hiroto Terashi, Shunsuke Sakakibara

**Affiliations:** 1Department of Plastic Surgery, Kobe University Graduate School of Medicine, Kobe 650-0017, Japan; terashi@med.kobe-u.ac.jp (H.T.); shunsuke@med.kobe-u.ac.jp (S.S.); 2Department of Oral and Maxillofacial Surgery, Kobe University Graduate School of Medicine, Kobe 650-0017, Japan; akashim@med.kobe-u.ac.jp

**Keywords:** *αSMA*, blue light, OPN3, human skin fibroblast, hypertrophic scar, peripheral circadian rhythm

## Abstract

It is unclear whether normal human skin tissue or abnormal scarring are photoreceptive. Therefore, this study investigated photosensitivity in normal skin tissue and hypertrophic scars. The expression of opsins, which are photoreceptor proteins, in normal dermal fibroblasts (NDFs) and hypertrophic scar fibroblasts (HSFs) was examined. After exposure to blue light (BL), changes in the expression levels of *αSMA* and clock-related genes, specifically *PER2* and *BMAL1*, were examined in both fibroblast types. Opsins were expressed in both fibroblast types, with OPN3 exhibiting the highest expression levels. After peripheral circadian rhythm disruption, BL induced rhythm formation in NDFs. In contrast, although HSFs showed changes in clock-related gene expression levels, no distinct rhythm formation was observed. The expression level of *αSMA* was significantly higher in HSFs and decreased to the same level as that in NDFs upon BL exposure. When OPN3 knocked-down HSFs were exposed to BL, the reduction in *αSMA* expression was inhibited. This study showed that BL exposure directly triggers peripheral circadian synchronization in NDFs but not in HSFs. OPN3-mediated BL exposure inhibited HSFs. Although the current results did not elucidate the relationship between peripheral circadian rhythms and hypertrophic scars, they show that BL can be applied for the prevention and treatment of hypertrophic scars and keloids.

## 1. Introduction

As the largest organ of the human body, the skin is perpetually exposed to external light and is susceptible to external stimuli. During the wound healing of skin tissue, particularly in dermal wounds, the process is categorized into the inflammation, proliferation, and remodeling phases. Following the inflammation phase, fibroblasts play a pivotal role in wound healing [1]. Fibroblasts produce the extracellular matrix (ECM) and undergo a transition to myofibroblasts, leading to wound contraction. During the remodeling phase, ECM degradation occurs, and myofibroblasts undergo apoptosis, resulting in mature scar formation [2,3]. However, aberrant healing processes, such as persistent inflammation, can induce abnormal scarring, including hypertrophic scars and keloids, attributable to the accumulation of myofibroblasts that escape apoptosis and to excessive ECM deposition [4].

The use of phototherapy for skin disorders has been reported [5] and is currently being applied in clinical settings [6]. Some reports suggesting the use of photoreception have also proposed that light exposure can promote wound healing (increasing the secretion of growth factors with red and green light) [7] and induce hair growth (accelerating the cell cycle with red light) [8]. However, the mechanisms underlying these effects are not fully understood [9].

In humans, opsins (OPNs), a type of photoreceptive protein, are well-known for their role in the reception of light. There are five known types: OPN1 (cone opsin, short wave: SW; middle wave: MW; long wave: LW), OPN2 (rhodopsin), OPN3 (encephalopsin/panopsin), OPN4 (melanopsin), and OPN5 (neuropsin). Each has its own sensitive wavelength range, and although there is some variation in reports, the peak sensitivities are generally considered to be OPN1 M/LW: 510–570 nm (red to green light); OPN1 SW: 420 nm (blue light); OPN2: 500 nm (red light); OPN3: 460–470 nm (blue light); OPN4: 460–480 nm (blue light); and OPN5: 360–380 nm (ultraviolet light) [10,11]. Opsins are G protein-coupled receptors, with OPN1 and OPN2 coupling with Gt, OPN3 with Gi/o, OPN4 with Gq, and OPN5 with Gi. Upon light reception, they convert the signal into an electrical signal through a second messenger, the G protein, thus activating their respective signaling pathways [10,11,12,13,14]. Human skin tissues express all opsin types [10,15]; however, their functions are not yet fully understood. In skin tissues, OPN3 demonstrates the highest reported expression among these OPN types [13,14].

Light entering through the eye, particularly blue light (BL), is transmitted as an electrical signal via OPN4 in the retina to the suprachiasmatic nucleus (SCN), primarily associated with the circadian rhythm [16,17,18,19]. Additionally, OPN3 and OPN5 are also involved in the central circadian rhythm [12]. Peripheral tissues, including the skin, are reported to possess their own circadian rhythms [20,21]. In mammals, the SCN regulates the independent rhythms of peripheral tissues throughout the body, acting as a master clock [22]. Corticosteroids are considered one of the synchronizing signals between the SCN and peripheral tissues [23]. In vitro studies have shown that dexamethasone (Dex) can reset peripheral circadian rhythms [24]. Interestingly, the local circadian rhythm of skin cells also affects wound healing and the hair cycle [25,26], and the dysregulation of peripheral circadian rhythms has been implicated in various skin disorders [27]. Nevertheless, the underlying photoreceptive mechanisms have not been clarified.

To date, light-induced synchronization of peripheral circadian rhythms has been reported only in mice [28]. Only one clinical study has investigated the effect of light exposure on scar tissue, using infrared light to do so, but the mechanism remains unclear [29]. Furthermore, the mechanisms involved in phototherapy from the perspective of synchronizing peripheral circadian rhythms have not been investigated yet. Therefore, this study aimed to confirm the synchronization of peripheral circadian rhythms in normal dermal fibroblasts (NDFs) and hypertrophic scar fibroblasts (HSFs), focusing on BL, which is known to influence circadian rhythms, and on OPN3, which has the highest expression in the skin compared to other types of opsins. Additionally, this study investigated the effects of BL on HSFs, exposing the relationship between photosensitive substances and peripheral circadian rhythms. This study aims to enable the development of novel phototherapeutic approaches for abnormal wound healing, such as hypertrophic scars.

## 2. Results

### 2.1. Cellular Characteristics

In NDFs, the expression of the alpha-smooth muscle actin (*αSMA*) gene increased upon treatment with transforming growth factor-β1 (TGF-β1) at 10 ng/mL. Furthermore, the expression levels of *αSMA* in fibroblasts derived from hypertrophic scars (HSFs) and NDFs treated with TGF-β were comparable (Appendix A). This suggested that HSFs predominantly included myofibroblasts.

### 2.2. Expression of Opsin Types

Quantitative analysis of opsin (OPN) types in each fibroblast revealed that opsin-3 (OPN3) was the most expressed (Figure 1a,b). Consequently, this study focused on OPN3 as the target photoreceptive protein. The expression of *OPN3* was confirmed through real-time quantitative reverse transcription PCR (qRT-PCR) using total RNA extracted from both fibroblast types and via the subsequent sequencing analysis of the PCR products (Appendix A).

### 2.3. Peripheral Circadian Rhythm Formation by BL

We assumed that OPN3 in the human skin tissue may also be involved in its circadian rhythm and examined time-dependent changes in the expression levels of clock-related genes as a preliminary study. Rhythm formation of the expression of either *BMAL1* or *PER2* was not observed after 6 d of culture under dark conditions for NDFs (*BMAL1*: *p* = 0.527; *PER2*: *p* = 0.116). As the positive control, rhythm formation was observed after the mRNA expression of both *BMAL1* and *PER2* was increased by Dex treatment (*BMAL1*: *p* = 0.020; *PER2*: *p* < 0.001) (Figure 2a). Upon BL irradiation, the mRNA expression of *PER2* in NDFs increased instantaneously; however, the mRNA expression of *BMAL1* did not increase instantaneously, although rhythm formation was observed in both (*BMAL1*: *p* < 0.001; *PER2*: *p* < 0.001) with a phase difference of about 11 h (Figure 2b). In HSFs, similar to NDFs, the rhythm formation of both *BMAL1* and *PER2* mRNA expression was absent after 6 d of culture under dark conditions (*BMAL1*: *p* = 0.947; *PER2*: *p* = 0.982). BL irradiation resulted in time-dependent changes in the mRNA expression of both *BMAL1* and *PER2*, with an approximate rhythmic change, but the variations were large and no statistically significant rhythm formation was observed (*BMAL1*: *p* = 0.429; *PER2*: *p* = 0.075) (Figure 2c).

### 2.4. Changes in αSMA Expression and the Role of OPN3 following BL Irradiation

Upon *OPN3* knockdown, a reduction in gene expression was confirmed in both fibroblast types (Figure 3a). Under baseline conditions, the expression level of *αSMA* was significantly higher in HSFs than in NDFs (*p* = 0.003). The expression level of *αSMA* in HSFs significantly decreased upon BL irradiation, reaching levels comparable to that in NDFs. Additionally, in HSFs, *αSMA* expression increased following BL irradiation after *OPN3* knockdown (*p* = 0.025). In contrast, in NDFs, *αSMA* expression did not change significantly with BL irradiation, or upon *OPN3* knockdown following BL irradiation (*p* = 0.101) (Figure 3b).

The expression levels of circadian-related genes were significantly higher in NDFs than in HSFs. In both NDFs and HSFs, there was no statistically significant difference in the expression levels of *BMAL1* and *PER2* 24 h after BL irradiation, compared to those before BL irradiation. Similarly, the knockdown of *OPN3* and BL irradiation showed little change (NDF: *BMAL1* [*p* = 0.669], *PER2* [*p* = 0.862]; HSF: *BMAL1* [*p* = 0.072], *PER2* [*p* = 0.407]) (Figure 3c,d).

### 2.5. Cell Viability

Cell viability significantly increased in NDFs upon BL irradiation (*p* = 0.017). However, no significant changes in cell viability were observed in NDFs with *OPN3* knockdown after BL irradiation (*p* = 0.345) (Figure 4a). In contrast, cell viability significantly decreased in HSFs upon BL irradiation, indicating cytotoxicity (*p* < 0.001). Additionally, cell viability significantly increased in HSFs with *OPN3* knockdown after BL irradiation (*p* < 0.001) (Figure 4b).

## 3. Discussion

This study revealed four key findings. First, the photoreceptive protein OPN3 was predominantly expressed in NDFs and HSFs. Second, BL induced the synchronization of the peripheral circadian rhythm in NDFs, but no rhythmic formation was observed in HSFs. Third, BL irradiation inhibited *αSMA* expression and cell viability in HSFs by modulating *OPN3* expression. Lastly, under conditions where the circadian rhythm was already established, BL irradiation exerted minimal impact on the expression of circadian-related genes.

In NDFs, the circadian rhythm, which disappeared from the cell cluster under dark conditions, was recovered via BL irradiation. In the peripheral circadian rhythm, the rhythm as a group was lost after at least 6 d under dark conditions, consistent with previous reports [30]. However, BL irradiation synchronized the peripheral circadian rhythm in NDFs, with *BMAL1* and *PER2* showing an approximately 11 h phase shift, as reported previously [31]. To the best of our knowledge, the present study is the first to scientifically demonstrate that the human skin is photoreceptive and related to the formation of the peripheral circadian rhythm. A previous study has shown that the circadian rhythm can be detected in non-neuronal cells as early as 1 h after the addition of Dex [32]; a similar result was obtained in our study with Dex treatment. Corticosteroids are regarded as the synchronizing signal between the SCN and peripheral tissues [26]. Based on the fact that the peripheral clock directly formed some rhythms after BL irradiation, we suggest that the rhythm of the skin (i.e., the peripheral clock) may be subject to dual control by both the central clock and light from the environment. Nevertheless, it is important to note that there are numerous other potential sources of stimulation as well [33]. Additionally, just as the peripheral clock in the skin reportedly associates with cell division as a way of avoiding DNA damage from ultraviolet rays [34], it also acts as a mechanism to respond flexibly to the environment. In contrast, HSFs exhibit changes in *BMAL1* and *PER2* expression after BL irradiation over time, but no clear rhythm formation is observed, potentially owing to pathological conditions. Moreover, aberrations in the peripheral circadian rhythm have been reported in relation to the pathology and progression of skin disorders like psoriasis and atopic dermatitis [35,36].

In this study, BL irradiation in HSFs reduced the expression of *αSMA*, whereas *OPN3* knockdown after BL irradiation inhibited this reduction. BL irradiation in HSFs also suggested cytotoxicity because cell viability decreased. However, cell viability increased after BL irradiation was followed by *OPN3* knockdown. This suggests that BL inhibits abnormal cells, with the involvement of OPN3 in the process. Reports that BL irradiation inhibits cancer-associated fibroblasts and fibrosis support our inference [37,38]. Moreover, BL irradiation inhibits the differentiation of skin fibroblasts into myofibroblasts induced by TGF-β1 [39,40]. Our results suggest that BL irradiation exerts an inhibitory effect on myofibroblasts even post-differentiation, indicating potential therapeutic effects in fibrotic skin diseases like hypertrophic scars.

In cultured cells, the expression of OPN3 was the highest among all opsins, as previously reported [13,14], with OPN4 expression being approximately one-tenth of OPN3 expression. However, in skin tissue, *OPN3* and *OPN4* expression levels were similar (Appendix A). Previously, we reported the expression of OPN4 in normal skin tissue and its involvement in BL reception in skin fibroblasts, a process leading to increased intracellular calcium influx and enhanced phosphorylation of ERK1/2 [15]. Furthermore, BL irradiation promotes the proliferation of skin fibroblasts [41,42]. Our results suggested that the inhibition of abnormal cells by BL is mediated by OPN3. OPN3 expressed in the skin is involved in maintaining homeostasis [13,43]. Thus, it can be inferred that BL reception via OPN4 promotes cell viability, while reception via OPN3 inhibits it, with both contributing to the maintenance of homeostasis in skin tissue.

This study also found an association between peripheral circadian rhythms and BL, but not to the extent of an associated functional relationship. The degree of BL used in this study might have been insufficient to alter the peripheral circadian rhythm under existing rhythmically established conditions. However, given that *BMAL1* is involved in fibrosis [44,45,46], our results, revealing differences in rhythm and clock-related gene expression levels between normal and scar fibroblasts, suggest a correlation between the peripheral circadian rhythm and skin disorders, warranting further investigation.

Whether BL is harmful to skin cells is controversial [47]. The dose of BL used in our study was lower than that used in many reports. We used approximately a dose of 1/100 to 1/1000 [9,47], and about a 1/10 dose of the BL component of sunlight (7.7 mW/cm^2^) [48]. In our results, BL did not show any particularly cytotoxic effects on NDFs, suggesting, as in previous studies, that higher doses might be more cytotoxic [49]. The dose used in our study appears to be relatively safe. However, in HSFs, even at low doses, BL suppressed *αSMA* and also showed cytotoxic responses. On the other hand, knockdown of OPN3 alleviated these effects, suggesting that even at low doses BL might act through OPN3 to exert inhibitory effects on abnormal cells.

There were some limitations to our study. First, this study evaluated gene expression levels, not protein levels. Although large differences among gene and protein expression levels are not expected, discrepancies between gene and protein expression levels are possible. We plan on investigating protein levels in the future. Second, the evaluations were conducted using cultured cells, and as expected the results were slightly different from the gene expression levels of opsins in actual tissues, suggesting possible phenotypic changes. Furthermore, in clinical settings, the question arises as to whether BL actually penetrates skin fibroblasts. It has been reported that BL penetrates the skin to a depth of about 0.5 to 1 mm (dermis) [50,51]. Therefore, we plan to conduct animal experiments and clinical research in the future. Third, the mechanism of entrainment of the peripheral circadian rhythm by BL irradiation in NDFs has not been clarified. We assume that opsins such as OPN3 or OPN4 are involved and are currently investigating possible associations. Lastly, the complex signaling mechanisms post-photoreception were not fully elucidated in this study. In HSFs, OPN3-mediated pathways, possibly involving the Gi-cascade as a second messenger, are indicated. This remains a subject for future investigation.

## 4. Materials and Methods

### 4.1. Primary Cell Culture

Normal dermal fibroblasts and HSFs were isolated from healthy skin tissue and hypertrophic scar tissue samples, respectively. These samples were obtained during diagnostic and therapeutic procedures. Sample collection was approved by the Kobe University Graduate School of Medical Research, Department of Medical Ethics Committee (approval number 1207), and conformed to the guidelines of the Declaration of Helsinki. Informed consent was obtained from all subjects.

After collection, the human skin tissue samples (including hypertrophic scar samples) were immediately cut into approximately 10 mm sections and immersed in 0.3% trypsin/PBS at 4 °C, overnight. These sections were further cut into approximately 3 mm sections, and the dermal side was placed on a culture dish and soaked in Dulbecco’s modified Eagle’s medium (DMEM; Wako, Osaka, Japan) supplemented with 10% fetal bovine serum (FBS). Once fibroblasts had grown and adhered, the skin tissue pieces were removed. For subculturing, the cells were maintained under subconfluent conditions, and Accutase (Nacalai Tesque, Kyoto, Japan) was used for cell detachment. Cultured cells from passage number 3 were utilized for the experiments.

Myofibroblasts involving hypertrophic scars were characterized via *αSMA* expression. Since the differentiation of normal fibroblasts into myofibroblasts was induced by TGF-β [52], when NDFs reached confluence, the medium was changed to serum-free DMEM and they were treated with 10 ng/mL TGF-β1 (rhTGF-β1, Fujifilm Wako, Osaka, Japan). After 24 h, total RNA was extracted and *αSMA* expression levels were compared with those in HSFs.

### 4.2. Total RNA Extraction

Total RNA was semi-automatically extracted from cultured cells using the Maxwell RSC Instrument (Promega, Madison, WI, USA) and the Maxwell RSC simplyRNA kit (AS1340, Promega), following the manufacturer’s instructions. Briefly, the cells were detached using Accutase (Nacalai Tesque) and collected via centrifugation at 1000 rpm for 10 min; the supernatant was discarded. The cell pellet was resuspended in a homogenization solution, followed by the addition of lysis buffer. Finally, total RNA was eluted in 50 μL RNase-free water.

### 4.3. qRT-PCR

Quantitative analysis of the mRNA expression of opsins was performed via qRT-PCR, using total RNA samples recovered from skin fibroblast primary cultures. β-Actin was used as an endogenous control. The qRT-PCR results were analyzed using the ΔΔC_t_ method along with TaqMan probes, performed in accordance with the protocol for the One Step PrimeScript RT-PCR Kit (Perfect Real Time) (Takara Bio, Kusatsu, Japan). Briefly, qRT-PCR was performed by mixing enzymes (including reverse transcriptases, Hot Start Taq DNA polymerase, and RNase inhibitors), a buffer (including a dNTP mixture and Mg^2+^), probes (5 μM), a forward primer (10 μM), a reverse primer (10 μM), total RNA, and RNase-free water. The PCR (Takara PCR Thermal Cycler Dice Version III TP600/Tp650) conditions were as follows: reverse transcription at 42 °C for 5 min and thermal denaturation at 95 °C for 10 s (one cycle), followed by thermal denaturation at 95 °C for 5 s, and annealing and extension at 60 °C for 30 s (40 cycles).

The primers were designed using the Perfect Real Time Primer Support System (Takara Bio). The primers are listed in Table 1.

### 4.4. Circadian Rhythm of NDFs and HSFs

NDFs and HSFs were seeded at 5.0 × 10^4^ cells per well, respectively, and 9-cis retinal (1 µM) was added before culturing them for 6 d at 37 °C and 5% CO_2_ without medium exchange to avoid light exposure. Total RNA was extracted every 6 h until 24 h. Next, 5.0 × 10^4^ fibroblasts were cultured for 7 d at 37 °C and 5% CO_2_ without medium exchange to avoid light exposure. After BL (450 nm) irradiation for 10 min or Dex (100 nM) addition (positive control), the cells were further cultured under dark conditions. Total RNA was extracted immediately after BL irradiation for 10 min or Dex treatment for 30 min, and then every 6 h till 24 h post-treatment. Circadian-related gene (*BMAL1* and *PER2*) expression levels were evaluated via qRT-PCR. The circadian rhythm reportedly exists in peripheral tissues such as the human skin [20], but almost disappears from cultures of primary cells isolated from mammalian tissues (e.g., liver, lung) within 2 to 7 d [30]. In this study, NDFs and HSFs were initially cultured for 7 d in the dark and then stimulated with Dex or BL (450 nm, 10 min). The mRNA levels of the brain and muscle Arnt-like protein-1 (*BMAL1*) and period2 (*PER2*) genes were quantitatively analyzed. This was performed immediately after the addition of the substances or irradiation, and subsequently at intervals of 6 h.

### 4.5. Irradiation

NDFs and HSFs (1.0 × 10^5^ cells per well, respectively) were seeded in 100 mm dishes and cultured at 37 °C under 5% CO_2_ in DMEM with 10% FBS. The experimental procedures, from the initial medium exchange to total RNA extraction, were conducted in a fully anechoic chamber under red lights. Once the cells reached subconfluence, BL was applied using LEDs (ISL-150 × 150 RB45, CCS Inc., Kyoto, Japan) from the overside of the culture dish from a distance of 10 cm above the culture dish (Appendix A). For BL irradiation (wavelength: 450 nm, 8.0 W/m^2^ [irradiance]), photon flux density was calculated by measuring the irradiance of the illumination from the LED device and was configured to 30 μmol m^−2^ s^−1^. BL irradiation was performed for 10 min at 37 °C [15]. This corresponded to a fluence of 0.5 J/cm^2^. After BL irradiation, the expression of the target genes (*OPN3*, *αSMA*) was evaluated using qRT-PCR.

### 4.6. siRNA Transfection

Fibroblasts were seeded and at 70% confluency, the medium was changed to serum-free DMEM. The fibroblasts were transfected with *OPN3* siRNA (Silencer Select siRNA, s24172, Thermo Fisher Scientific, Waltham, MA, USA) using Lipofectamine RNAiMAX Transfection Reagent (Invitrogen/Thermo Fisher Scientific) [53], and 9-cis retinal (1 μM) was added. After 2 d of incubation, the cells were exposed to BL (450 nm) for 10 min in a dark room. Total RNA was extracted 24 h post-irradiation.

### 4.7. Cell Viability

NDFs and HSFs (1.0 × 10^4^ cells per well, respectively) were seeded in 96-well plates and incubated at 37 °C and 5% CO_2_. After medium change the following day, the cells were cultured to subconfluence. After 24 h dark adaptation, the cells were exposed to BL for 10 min; the control group did not receive BL irradiation. Cell proliferation and cytotoxicity were assessed using the Cell Counting Kit-8 (Dojin Institute for Chemical Research, Kumamoto, Japan).

### 4.8. Statistical Analysis

Data are presented as mean ± standard deviation of the mean. Statistical analyses were performed using the R software, version 4.2.1 (R Foundation for Statistical Computing, Vienna, Austria).

For comparisons involving two groups, the Student’s *t*-test was conducted. For comparisons of three or more groups, one-way analysis of variance was performed, and Tukey’s test was used for post hoc verification. The circadian rhythm was analyzed using the formula: y = interception + amplitude × cos (2πt/D − acrophase) + ε (t: time variable; D: fixed period; *ε*: an error term with mean 0), following the cosinor method (using the “cosinor2” version 0.2.1 package in R). The significance level was set at 0.05.

## 5. Conclusions

This study demonstrated the expression of opsins in normal skin fibroblasts and fibroblasts derived from hypertrophic scars, with OPN3 being the most prominently expressed. Both of these cell types exhibited sensitivity to BL. In NDFs, it was possible to synchronize the disrupted peripheral circadian rhythms. In HSFs, the peripheral circadian rhythms appeared to be irregular. Furthermore, although the findings of this study do not conclusively establish a relationship between peripheral circadian rhythms and hypertrophic scars, the collective evidence from this research and prior studies, wherein BL irradiation has been observed to inhibit the differentiation of skin fibroblasts into myofibroblasts, suggests the potential utility of BL not only in the prevention, but also in the treatment of abnormal skin scarring.

## Figures and Tables

**Figure 1 ijms-25-02126-f001:**
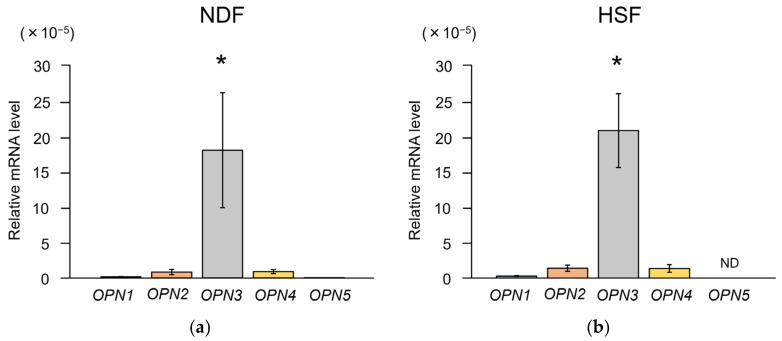
Gene expression levels of opsins in fibroblasts derived from normal skin tissue and hypertrophic scars. (**a**) Normal dermal fibroblasts (NDFs) derived from normal skin tissue. The expression level of OPN3 was significantly higher than that of other opsins (n = 4; relative to the expression level of β-Actin; Tukey’s test, * *p* < 0.001). (**b**) Hypertrophic scar fibroblasts (HSFs) derived from hypertrophic scars. The expression level of opsin-3 (OPN3) was significantly higher than that of other opsins (n = 4; relative to the expression level of β-Actin; Tukey’s test, * *p* < 0.001). Opsin-5 was not detected (ND, not detectable).

**Figure 2 ijms-25-02126-f002:**
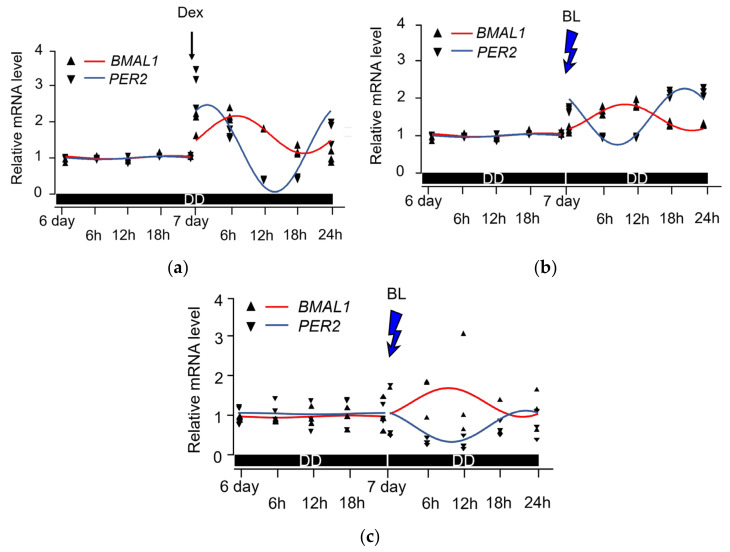
Impact of blue light (BL) on peripheral circadian rhythms in NDFs and HSFs. (**a**) In NDFs, real-time quantitative reverse transcription PCR (qRT-PCR) analysis of brain and muscle Arnt-like protein-1 (*BMAL1*) and period2 (*PER2*) mRNA reveals the disappearance of the rhythm after 6 d of culture without light (n = 3; day 7 value as the reference; cosinor method; *BMAL1*: *p* = 0.527; *PER2*: *p* = 0.116). Adding dexamethasone (Dex) on day 7 restored the rhythm for both genes (n = 3; day 7 value as the reference; cosinor method; *BMAL1*: *p* = 0.015; *PER2*: *p* < 0.001). DD, constant darkness. (**b**) In NDFs, qRT-PCR of *BMAL1* and *PER2* mRNA shows that BL irradiation forms the rhythm for restoring clock-related genes (n = 3; day 7 value as the reference; cosinor method; *BMAL1*, *PER2*: *p* < 0.001). (**c**) In HSFs, qRT-PCR analysis of the *BMAL1* and *PER2* mRNA reveals the disappearance of the rhythm after 6 d of culture without light as well (n = 3; day 7 value as the reference; cosinor method; *BMAL1*: *p* = 0.947; *PER2*: *p* = 0.982). Despite BL irradiation, considerable variability was observed at each time point, and no statistically significant rhythm formation was observed (n = 3; day 7 value as the reference; cosinor method; *BMAL1*: *p* = 0.429; *PER2*: *p* = 0.075).

**Figure 3 ijms-25-02126-f003:**
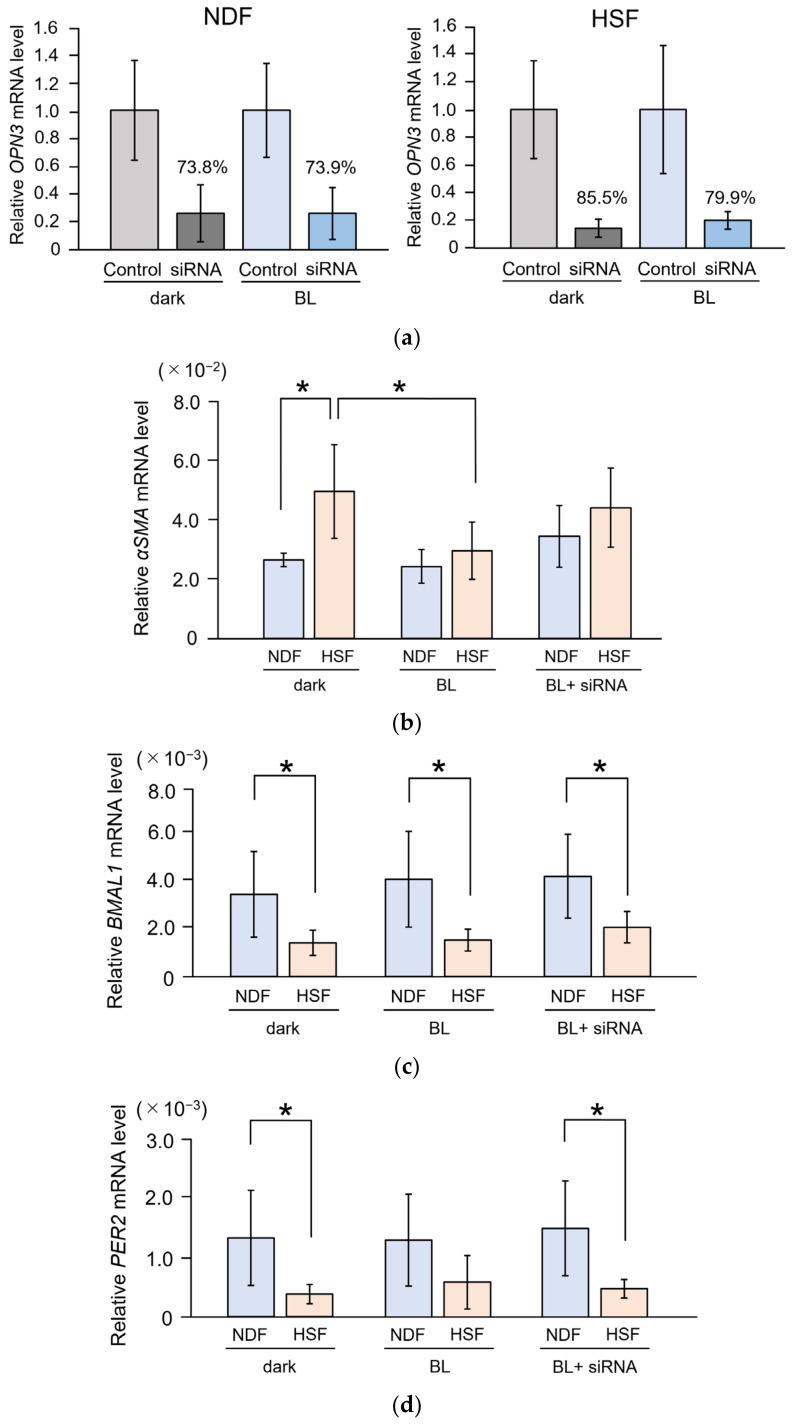
Effect of BL and OPN3 on fibroblasts derived from normal skin tissue and hypertrophic scars. (**a**) Knockdown of *OPN3* in NDFs and HSFs. The knockdown efficacy was >70%. (**b**) The impact of BL on *αSMA* expression in NDFs and HSFs. The expression level of *αSMA* was significantly higher in HSFs compared to that in NDFs (n = 7; Student’s t-test; *p* = 0.003; * *p* < 0.05). BL irradiation significantly reduced *αSMA* expression in HSFs (n = 7; Tukey’s test; *p* = 0.025; * *p* < 0.05). The reduction in *αSMA* expression caused by BL irradiation was not statistically significant following *OPN3* knockdown (siRNA) (n = 7; Tukey’s test; *p* = 0.725). (**c**) The effect of BL on *BMAL1* expression in NDFs and HSFs. BMAL1 expression was significantly higher in NDFs compared to that in HSFs (n = 7; Student’s t-test; *p* = 0.014, *p* = 0.006, and *p* = 0.011 under dark, BL, and BL+ siRNA conditions, respectively; * *p* < 0.05). No significant changes in expression were observed following BL irradiation (n = 7; analysis of variance; NDFs, *p* = 0.669; HSFs, *p* = 0.072). (**d**) The effect of BL on *PER2* expression in NDFs and HSFs. *PER2* expression was significantly higher in NDFs compared to that in HSFs, under dark conditions, and BL irradiation following *OPN3* knockdown (siRNA) (n = 7; Student’s t-test; *p* = 0.010, *p* = 0.059, and *p* = 0.006 under dark, BL, and BL+ siRNA conditions, respectively; * *p* < 0.05). No significant changes in expression were observed following BL irradiation (n = 7; analysis of variance; NDF, *p* = 0.862; HSF, *p* = 0.407).

**Figure 4 ijms-25-02126-f004:**
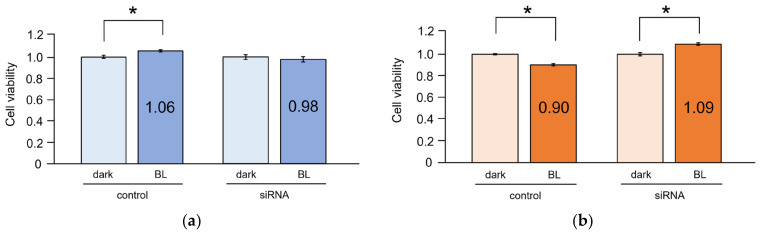
Impact of BL on cell viability in fibroblasts derived from normal skin tissue and hypertrophic scars. (**a**) Effect of BL on cell viability in NDFs. BL irradiation significantly increased cell proliferation in NDFs (n = 7; dark condition is used as the reference; Student’s *t*-test; *p* = 0.017; * *p* < 0.05). Knockdown of *OPN3* (siRNA; efficacy: 90.3%) followed by BL irradiation did not show any increase in cell proliferation (n = 7; dark condition is used as the reference; Student’s t-test; *p* = 0.345). (**b**) Effect of BL on cell viability in HSFs. BL irradiation significantly reduced cell proliferation in HSFs (n = 7; dark condition is used as the reference; Student’s t-test; *p* < 0.001; * *p* < 0.05). Knockdown of *OPN3* (siRNA; efficacy: 74.3%) followed by BL irradiation resulted in a significant increase in cell proliferation (n = 7; dark condition is used as the reference; Student’s t-test; *p* < 0.001; * *p* < 0.05).

**Table 1 ijms-25-02126-t001:** List of real time PCR primers and probes used in this study.

Primer	Forward (5′-3′)	Reverse (5′-3′)	Probe (5′-3′)
*OPN1*	GGCCCTGAAAGCTGTTGCA	GCACGTAGCAGACACAGAAGG	ATCACAACCACCATGCGGCTCACCTC
*OPN2*	CCGTCAAGGAGGCCGCTG	CACCCAGCAGATCAGGAAAGC	CAGCAGCAGGAGTCAGCCACCACAC
*OPN3*	GGCAGCCTCTTCGGGATTG	CACTCTGGCATGGACCACG	TTCCATTGCCACCCTAACCGTGCTGG
*OPN4*	CCCCTGTCTTCTTCACCAGT	GATTACCAGGTAGCGGTCCA	ATAGAACTCGCAGCCTGTCTCCCCAAA
*OPN5*	CTGCAGCGATGTACAATCCC	GCACAGCAGAAGACTTCCTG	TGCAGCCTGAAGCCTTCCAGAGACTT
*αSMA*	TCCAGGCGGTGCTGTCTC	CTCGGCCAGCCAGATCCA	CCTCTGGACGCACAACTGGCATCGTG
*Per2*	GGACAGCGTCATCAGGTACTTG	CCGCTTATCACTGGACCTTAGC	CTCGCATTTCCTCTTCAGGGTGGCAGC
*Bmal1*	CACCAATCCATACACAGAAGCAA	CTTCCCTCGGTCACATCCTAC	TGAAACACCTCATTCTCAGGGCAGCAGATG
*β-Actin*	TTGGCAATGAGCGGTTCCG	GGAGTTGAAGGTAGTTTCGTGGA	CCTTCCTTCCTGGGCATGGAGTCCTGTG

## Data Availability

The data presented in this study are available on request from the corresponding author.

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
