# Peer review of "Differential Photosensitivity of Fibroblasts Obtained from Normal Skin and Hypertrophic Scar Tissues"

_ijms, 2024, doi:10.3390/ijms25042126_

Round 1

Reviewer 1 Report

Comments and Suggestions for Authors

The work entitled “Differential photosensitivity of fibroblasts obtained from normal skin and hypertrophic scar tissues” describes the behavior of fibroblasts and the expression of opsin 1, 2, 3, 4, and 5 in the presence of blue light. They identify OPN3 as the gene most responsive to the light signal.

On the other hand, they evaluate the expression of the BMAL1 and PER2 genes before light stimulation, where non-dermal fibroblasts (NDF) respond similarly with blue light as with dexamethasone induction, while hypertrophic fibroblasts derived from scars (HSF) They did not react again to the two stimuli. Likewise, they describe the dependence of ONP3 expression for the induction of α-SMA.

The article is interesting and challenging in proposing a peripheral circadian response of fibroblasts dependent on direct light; however, the same authors are aware that their experiments lack the determination of the proteins corresponding to the expression of the genes they measured. It is important to note that, despite the relevance of opsins in the work, their treatment in the introduction is particularly scarce.

Author Response

Thank you for the review of our manuscript. The manuscript has greatly benefitted due to the useful suggestions provided. We have sincerely taken your comments into consideration and have done our utmost to address the issues raised. Regarding Figure 2, we have made some changes due to the points raised about copyright. Please find bellow a point-by-point response to the comments. Revised sections in the text are highlighted in yellow.

Reviewer 2 Report

Comments and Suggestions for Authors

Introduction:

line 31- is not eXternal light? Please correct.

lines 47-54: is there any relations regarding the oxidative stress caused by blue light and the alterations explained at this paragraph? If yes, please include in the introduction.

Please include some information about dexamethasone in the introduction since it appears to be important for the further comprehension.

Results

lines 89-109: I felt the most of the information are material and methods. Please correct this huge paragraph.

Discussion

Since we are talking about blue light, I am interested to know about how this work can be comprehend by the utilization of eletronic devices and the effects on skin. Please clarify this point, perhaps explaining better the potency of blue light irradiation.

Material and methods

Please include information regarding how much blue light is irradiated (joule per area unit) and the distance. This is crucial to understand if this work is related with blue light from sun or eletronic devices.

Conclusions

During my evaluation it was not clear how "the potential application of BL in the prevention and treatment of skin abnormal scarring has been established". Based in which results can you have this conclusion? Please include a more detailed discussion and point in the conclusions which parameters were important to evaluate the BL for prevention and treatment of skin abnormal scarring.

Author Response

(The authors gave the same response as above.)

Reviewer 3 Report

Comments and Suggestions for Authors

Dear Authors,

I read with interest the article entitled: “Differential photosensitivity of fibroblasts obtained from normal skin and hypertrophic scar tissues”. In my view, it is a high quality study, not only for the study, but also for the possible clinical applications.

I believe the article is of interest and well-written. I would like to know why you assumed that Opsinas are involved in the circadian rhythm? And why did you decide to adress the photosensitivity of normal vs hyperthrophic fibroblast scars?

The article, the presentation and the results are good. I would like to point out the originality os this study, regarding and searching blue light impact on fibroblasts.

Introduction:

-Line 43: not all lights, red led light. The mechanism is described, please add it ( TfG.beta, ROS)

-Line 47: Why blue light? Justify. Blue light is nearly damage to the skin. Do you think that blue light penetrate to fibroblasts in the skin?

Results:

-Line 88: Why did you assumed that OPN3 is involved in the circadian rythm? This assertion need to be stated in the introduction.

-Line 101: Blue light irradiation, doses and irradiance need to be measured. You also need to describe what kind of dispositive did you used, arrays or whatever and how.

-Line 161: What doses were used? Blue light normaly decrease cell viability.

Discusion:

-Line 174: again light doses of blue light, are real? You did not state that. What are the normal doses received in the skin and do they penetrate in fibroblast? Maybe you would need to add some references.

-Line 190: I believe this affirmation is not completely true, as normaly what skin receive is all the spectrum, visible light. Blue light is the one which less penetration. Maybe you need to add some references, maybe blue light represented only a part of the stimuli.

-Line 221: In my view the light doses need to be discuss deeper, and stated

Best wishes,

Comments on the Quality of English Language

Dear Guest Editor,

In my opinion this is a very interesting manuscript as is testing blue light in fibrobast searchin a future clinical implication in medical practise.

It is very original and well written and presented.

Best wishes,

Author Response

(The authors gave the same response as above.)

Round 2

Reviewer 2 Report

Comments and Suggestions for Authors

The corrections are satisfactory. Congratulations for your work.